environmental chemistry/chemical engineering

HRF5015, cyclohexanone, self-condensation reaction, dimer, selectivity

**Author for correspondence:**
Guoxin Sun
e-mail: sun-guo-xin@hotmail.com

This article has been edited by the Royal Society of Chemistry, including the commissioning, peer review process and editorial aspects up to the point of acceptance.

# Highly selective self-condensation of cyclohexanone: the distinct catalytic behaviour of HRF5015

Xiujing Peng[1,2], Shah Zeb[1,2], Jianguo Zhao[1], Miaomiao Zhang[1], Yu Cui[1] and Guoxin Sun[1,2]

[1]School of Chemistry and Chemical Engineering, and [2]Institute for Smart Materials and Engineering, University of Jinan, 250022, Jinan, People's Republic of China

GS, 0000-0002-1081-1144

HRF5015, a perfluorosulfonic acid resin catalyst with unique pore structures, was investigated in the catalytic self-condensation of cyclohexanone under mild conditions. The morphology of HRF5015 was characterized by transmission electron microscopy (TEM) and atomic force microscope (AFM), and the reaction mechanism was studied by *in situ* diffuse reflectance infrared Fourier transform spectroscopy (DRIFTS). The effects of reaction time and temperature on the yield of dimer were investigated under the nitrogen atmosphere. The results show that the reaction temperature is low, and especially, the selectivity of the dimer is close to 100%. The apparent activation energy for the dimer formation reaction is $54 \, \text{kJ} \, \text{mol}^{-1}$. Synergistic action of cluster structure formed by sulfonic groups and nanopores in HRF5015 may be the key factor of high-efficiency catalytic activity and high selectivity. *In situ* infrared spectra indicate that the intermediate is stable in the reaction process. HRF5015 is environmentally friendly and re-usable, which shows good potential in a future application.

## 1. Introduction

Cyclohexanone is an important intermediate for synthesizing fine chemicals [1] and also regarded as a promising second-generation biofuel [2,3]. The dimers by the self-condensation of cyclohexanone include a pair of resonance structures of 2-(1-cyclohexenyl)cyclohexanone and 2-cyclohexylidenecyclohexanone, which can be readily dehydrogenated directly to o-phenylphenol (OPP) [4]. OPP is an important organic chemical

product, which is widely used in fungicides, flame retardants and other fields [5]. The key to obtaining OPP is to prepare cyclohexanone dimers with high selectivity and minimize the formation of trimers and polymers.

Cyclohexanone self-condensation is a reversible aldol condensation reaction that can be catalysed by acidic catalysts or alkaline catalysts [6,7]. Sulfuric acid is a widely used catalyst for this reaction in industry, but equipment corrosion and environmental pollution are still urgent problems to be solved.

Heterogeneous catalysts are easily separated from the reaction mixture by a simple physical process, thus providing a more environmentally friendly process [8]. Many researchers reported resins as catalysts for cyclohexanone self-condensation [9,10]. With Lewatite SPC118 W as the catalyst, the highest conversion of cyclohexanone reached 64%, while the selectivity reached 70%, resulting in the difficult separation of products [11]. The trimer yield was close to 10%, and the dimer yield was 75%, with Amberlyst 15 as catalyst under 100°C for 500 min [12]. Because the monocondensation product can react further with cyclohexanone to form trimer or multimer, the selectivity of self-aldol condensation should also be concerned for catalyst development [13]. It is highly desirable to find a suitable, environmentally friendly and heterogeneous solid acid catalyst with high selectivity and activity for the self-condensation reaction of cyclohexanone.

Perfluorosulfonic acid (PFSA) resin has special properties due to the presence of fluorine atoms. PFSA is widely used in the polymer electrolyte membrane of fuel cells [14,15]. The strong acidity of PFSA makes it an active catalyst for some organic reactions [16,17]. Compared with traditional catalysts, PFSA has many advantages, such as unique catalytic activity, environmental friendliness and re-usability [18].

The mass transfer and viscosity of the fluid in the nanochannel are different from that of the bulk phase, which significantly accelerates the reaction rate. The successful development and application of microchannel reactors are based on this point. In this work, HRF5015, a kind of perfluorosulfonic acid resin with a unique nanoporous structure, was used as a catalyst for the self-condensation of cyclohexanone. The nanoporous structure helps to increase the number of catalytic sites, and nanochannels provide a unique microenvironment for the reaction. The experimental results show that HRF5015 has high catalytic activity and selectivity for the selective self-condensation of cyclohexanone.

# 2. Materials and methods

## 2.1. Reactants and materials

HRF5015, with the acid concentration of 1.06 mmol g$^{-1}$, was produced by Shandong Huaxia Shenzhou New Material Co. Ltd. Cyclohexanone was bought from Aladdin Reagent Co. Ltd, and the purity of cyclohexanone was more than 99% analysed by high-performance liquid chromatography (HPLC).

## 2.2. Characterization

The pore structure of HRF5015 was characterized by transmission electron microscopy (TEM: JEOL-1200EX) and atomic force microscope (AFM: Bruker Dimension Icon). Cyclohexanone and self-condensation products were identified and quantified by GC/MS (GCMS-QP2020, 19091 J-413, 30 m × 320 µm × 0.25 µm). The inlet temperature was 200°C, and the oven and detector temperature were both 220°C. *In situ* diffuse reflectance FTIR spectra were recorded using a Bruker Tensor II spectrometer over the range 4000–400 cm$^{-1}$, with 16 scans at a resolution of 4 cm$^{-1}$. Diffuse reflection accessories were used for *in situ* infrared measurement.

## 2.3. Catalytic reaction

To prevent the oxidation of reactants, the catalytic reaction was carried out in a nitrogen atmosphere. A certain amount of cyclohexanone was added to the reaction vessel, and the calculated amount of catalyst was added when the temperature reached the set value. GC/MS was used to analyse the samples at different time intervals [12]. The yield of the dimer was calculated by the result of GC/MS.

# 3. Results

## 3.1. Catalytic performance investigation

The reaction temperature is an essential factor affecting the composition of chemical reaction products [19]. Figure 1 shows the yield of dimer versus time at different reaction temperatures. The yield of

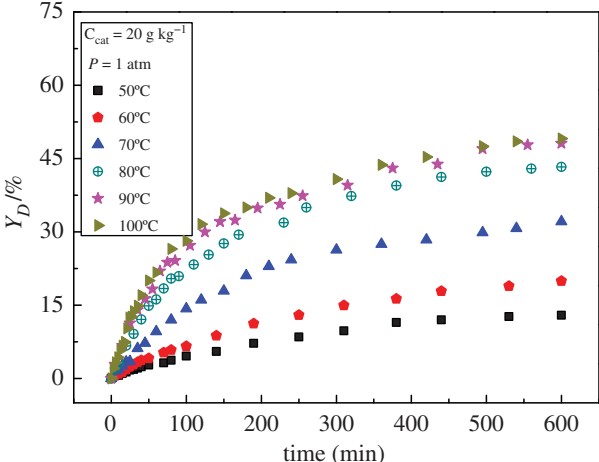

**Figure 1.** Effect of temperature on dimer yield.

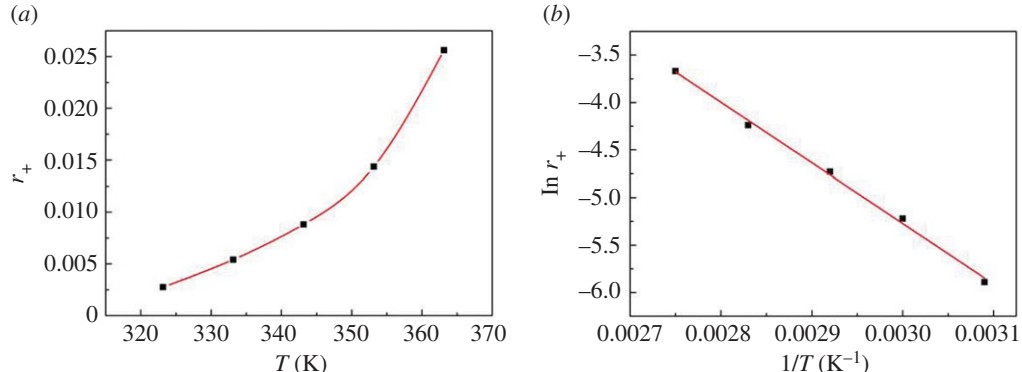

**Figure 2.** The relationship between temperature and forward reaction rate.

dimer increases from 14% to 40% when the reaction temperature rises from 50°C to 100°C within 250 min. The dimer yields do not have a noticeable difference at the reaction temperatures of 90°C and 100°C, which indicates that the yield of dimer will not increase significantly with the further increase of temperature.

In the initial stage of reaction, the yield increases rapidly. As the reaction progresses, the reverse reaction rate also increases, which leads to a gradual decrease in the overall reaction rate and a slowing growth rate of yield. The reaction can occur even at 50°C. The catalyst has excellent performance for this reaction. The initial forward rate ($r_+$) was calculated under different temperatures from the data of figure 1.

The reaction rate rises sharply in the range of temperature studied with the increase of temperature (figure 2). The activation energy was calculated by using the initial forward rate. The relationship between the initial forward rate and temperature is shown in figure 2. According to the Arrhenius equation [20], the value of apparent activation energy for the dimer formation from cyclohexanone was 54 kJ mol$^{-1}$. This value is lower than that reported in the literature [12,20], 68.46 and 177.1 kJ mol$^{-1}$, respectively, which suggests that the HRF5015 catalyst has better catalytic activity.

A significant indicator of product manufacturing is the selectivity of the reaction. From table 1, it can be seen that the products contain only dimers (2-(1-cyclohexenyl) cyclohexanone and 2-cyclohexylidencyclohexanone) under different reaction temperature. It did not find trimers and even tetramers in the process even up to 100°C. The selectivity of the dimer is near 100%, with HRF5015 as the catalyst for cyclohexanone self-condensation reaction. However, multimers appeared in the system catalysed by other catalysts [12].

The reaction conditions and results in the literature on cyclohexanone condensation reactions are summarized and listed in table 2.

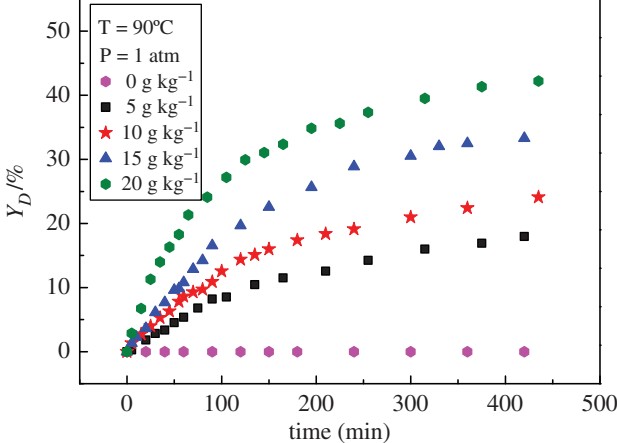

**Figure 3.** Effect of catalyst amount on dimer yield.

**Table 1.** Composition of the products at different temperatures.

| reaction temperature | composition analysis of reaction products (%) | | | | dimer selectivity (%) |
| --- | --- | --- | --- | --- | --- |
| | cyclohexanone | 2-(1-cyclohexenyl) cyclohexanone | 2-cyclohexylidene-cyclohexanone | trimer | |
| 50°C | 86.17 | 12.38 | 1.45 | 0.00 | 100 |
| 60°C | 79.51 | 18.36 | 2.13 | 0.00 | 100 |
| 70°C | 66.17 | 30.27 | 3.56 | 0.00 | 100 |
| 80°C | 56.23 | 39.17 | 4.60 | 0.00 | 100 |
| 90°C | 51.35 | 44.10 | 4.55 | 0.00 | 100 |
| 100°C | 49.73 | 45.33 | 4.94 | 0.00 | 100 |

**Table 2.** Experimental data for self-condensation of cyclohexanone.

| catalyst | reaction temperature | dimer selectivity | reference |
| --- | --- | --- | --- |
| HRF5015 | 90°C | approximately 100% | this work |
| Lewatite SPC118 W | 142°C | 70% | [11] |
| NaOH | 140°C | 94.5% | [7] |
| alkali carbonate | 130°C | 93% | [21] |
| $PW_{12} + PMo_{12}/SBA-15$ | 150°C | 90% | [22] |

As can be seen from table 2, with HRF5015 as catalyst for the cyclohexanone condensation reaction, the reaction temperature was lower than that of other catalysts. The selectivity of chemical reactions is crucial to production. The dimer selectivity can reach nearly 100% with HRF5015.

The amount of catalyst has a significant influence on the yield of the dimer. The amounts of catalyst were set to 0, 5, 10, 15 and 20 g kg$^{-1}$, respectively. The results in figure 3 show that the self-condensation reaction of cyclohexanone could not occur without catalyst at 90°C. With the increase of catalyst amount, the yield of dimer increased gradually. A large number of catalysts provide more catalytic reaction sites.

One of the most remarkable advantages of heterogeneous catalysts is that they can be easily removed and recovered from the reaction medium. The reaction temperature selected in this experiment is 50–100°C, and the catalyst can maintain long-term stability. Besides, it is insoluble in ketone compounds. HRF5015 exists in the form of solid particles before and after the reaction, so the reaction is heterogeneous. After the reaction, the catalyst HRF5015 was filtered and washed five times with ethanol to remove the reactant and

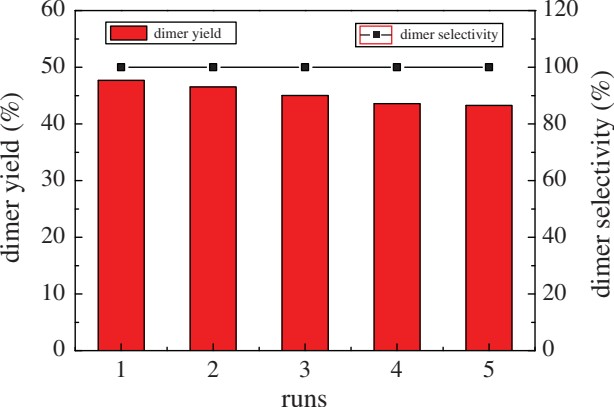

**Figure 4.** Catalyst durability test of HRF5015.

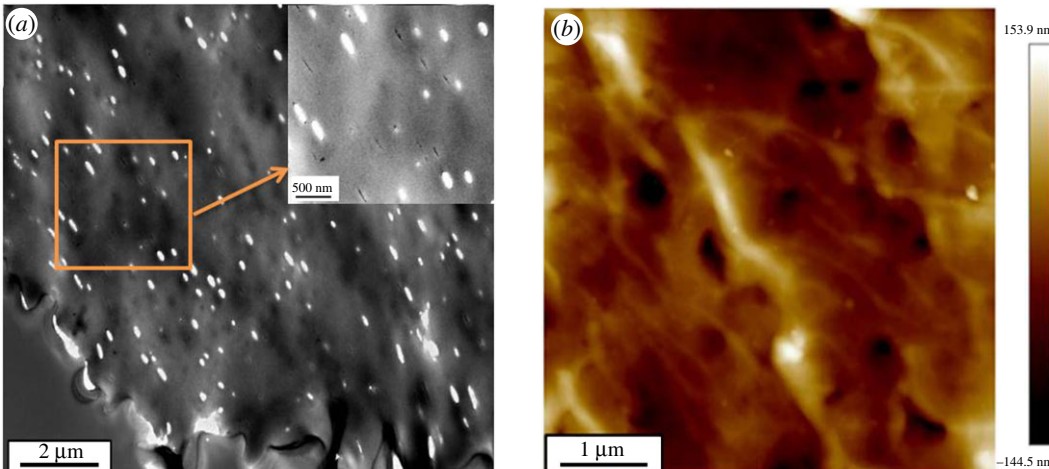

**Figure 5.** TEM image (*a*) and AFM image (*b*) of HRF5015.

products. Then, it was dried overnight at 60°C in a vacuum oven. The catalyst was re-used five times without a significant reduction of dimer yield and selectivity. Figure 4 shows that the selectivity of the dimer is almost 100%, which is superior to other catalysts reported at present, and there are no trimers and polymers [12].

## 3.2. Discussion of structure and catalytic properties

The catalysts were characterized by various means. HRF5015 was firstly embedded in epoxy resin and then sliced. The pore size and distribution of HRF5015 were observed by TEM. The result is shown in figure 5*a*.

It is evident that HRF5015 contains nanopores with different sizes between 100 and 300 nm. Figure 5*b* is the AFM image of the catalyst surface. The resin surface is rough. There are many penetrating nanopores with a depth of more than 150 nm. The above characteristics make it easy for the reactants to enter the inside of the catalyst particles, which is beneficial for accelerating the reaction rate.

The cluster-channel model is generally accepted in the study of perfluorinated ionic membrane structure [23]. This model suggests that the diameter of clusters formed by hydrophilic sulfonic acid groups is 4 nm, and the channel connecting clusters is 1 nm. In other words, the macroporous walls of the resin matrix are covered with such microstructures. The strong electronegativity of fluorine atoms makes the polymer material possess excellent chemical stability. At the same time, it can stabilize sulfonate anions and promote proton ionization through the electron absorption property of the polymer framework, which is conducive to accelerating the reaction kinetics and improving the acid strength [24]. In this structure model, high-density sulfonic acid groups are aggregated in the clusters, similarly to the nanoreactor, which further improves the catalytic efficiency. Another reason for the

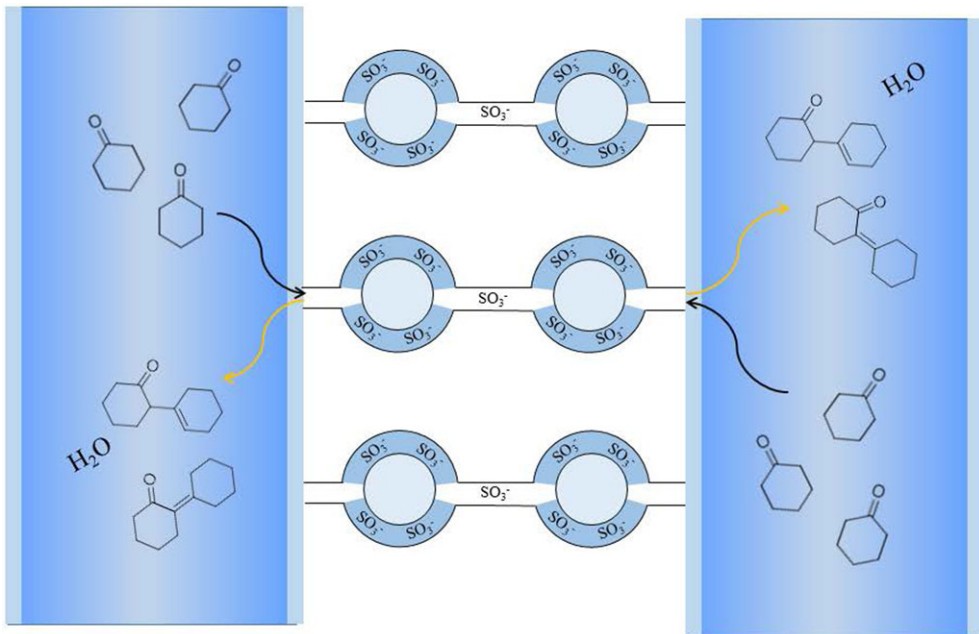

**Figure 6.** Reaction schematic in HRF5015 nanochannel.

high efficiency of the catalyst may be attributed to nanochannels. The large specific surface area of nanochannels exposes more acidic sites.

The friction coefficient decreases with the increase of hydrophobicity of the channel wall in rough nanochannels [25]. Because of the slip effect in nanochannels [26], the flow resistance of cyclohexanone in microchannels decreases significantly [27–29]. The decrease of pressure drop makes more cyclohexanone molecules enter the nanochannels and interact with sulfonic acid groups.

The influence of the interaction between the fluid molecules and the wall surface on the viscosity coefficient in the nanochannels cannot be ignored. The viscosity of the fluid in the nanochannel is not only directly related to the liquid, but also takes into account the influence of the nanoenvironment. From the perspective of energy, the deeper the surface potential trap is, the more chances it has to capture fluid molecules, and the shallower the potential well is, the fewer molecules it will capture. If the trap depth is small, the viscosity of the fluid in the nanochannel may be lower than that in the volume-phase system [30]. For the catalyst we used, because the inner wall of the nanopores is full of fluorine atoms, a weak interaction is speculated with the reactant and products. The potential well is shallow, which makes the products very easy to enter and remove quickly, resulting in the fast forward rate.

In this paper, a significant result is the very high catalytic selectivity. This cluster-channel model (figure 6) is suitable to explain the high selectivity of catalysts. We optimized the molecular conformation and analysed the molecular sizes of cyclohexanone, dimer and trimer (figure 7). The dimensions of cyclohexanone, product and trimer along the longest axis are 0.51, 0.84 and 1.16 nm, respectively. In the cluster-channel model, the diameter of nanochannel is 1 nm, only the product dimer can pass through the channel, but the trimers are difficult to get through. This special nano-microstructure [31] is probably the principal reason for the high selectivity of catalysts HRF5015.

## 3.3. In situ infrared study

HRF5015 particles were ground into powder in liquid nitrogen. The powder was laid flat in the diffuse reflection tank. It was heated slowly up to 50°C under a flow of $N_2$ (100 ml min$^{-1}$) to remove any adsorbed species. Then, the samples were cooled to 10°C after 5 min. Cyclohexanone is bubbling into the reaction chamber with nitrogen gas for 45 min. Then, the reaction tank was purged with nitrogen for 10 min. The infrared spectra of the samples were collected under 60°C, 70°C and 80°C, respectively.

Figure 8a showed that the peaks between 1700 and 1760 cm$^{-1}$ could be divided into three peaks, 1740, 1731 and 1717 cm$^{-1}$, respectively. The 1740 cm$^{-1}$ is assigned to the C=O vibration absorption peak for adsorbed cyclohexanone on the surface of the catalyst. With the increase of temperature, the peak of

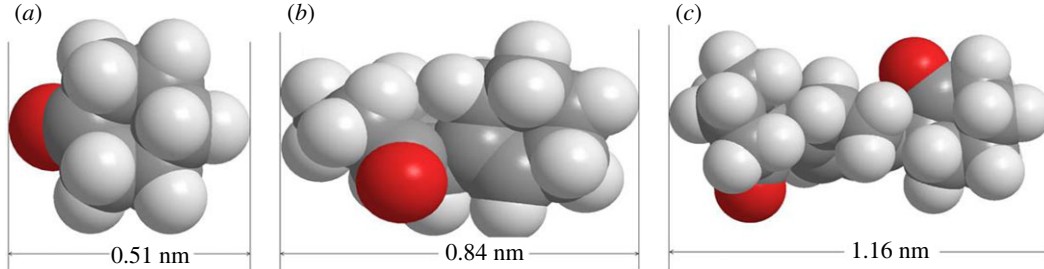

**Figure 7.** Maximum axis length of different molecules (*a*) cyclohexanone, (*b*) dimer and (*c*) trimer.

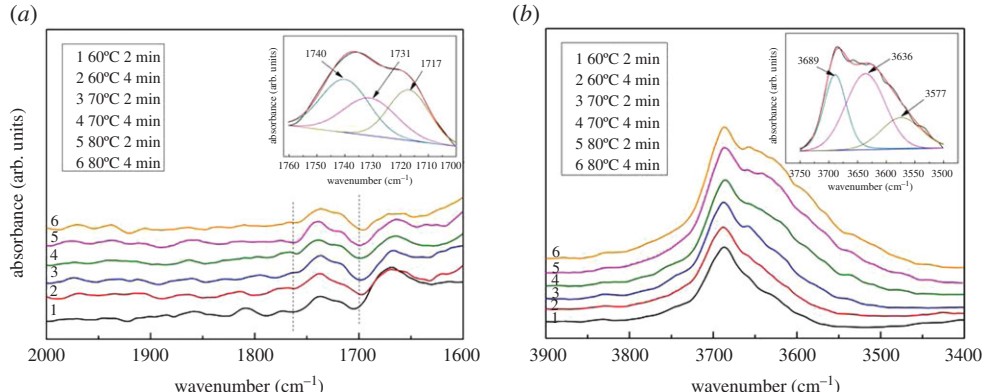

**Figure 8.** *In situ* infrared spectra of adsorbed cyclohexanone catalysed by HRF5015.

**Scheme 1.** Reaction mechanism of cyclohexanone catalysed by HRF5015.

$1731 \ cm^{-1}$ increases gradually, which is the C=O vibration absorption peak of product dimer (scheme 1), and $1717 \ cm^{-1}$ may be the C=O vibration absorption peak of intermediate. The absorption peaks in the range of $3500–3900 \ cm^{-1}$ were also analysed in figure 8*b*. It was found that the absorption peaks consisted of 3689, 3636 and $3577 \ cm^{-1}$. The peak at $3689 \ cm^{-1}$ is generated by cyclohexanone (scheme 1); $3636 \ cm^{-1}$ is the OH peak in the product water (scheme 1) and the intensity increases gradually, which is the same as $1731 \ cm^{-1}$; $3577 \ cm^{-1}$ should be the OH peak of the intermediate (scheme 1), similar to $1717 \ cm^{-1}$ peaks. According to the change of *in situ* infrared spectra, we speculated that the carbonyl group and the hydroxyl group of the intermediate had the same change trend, indicating that the intermediate was stable in the reaction process. The reaction mechanism [7] in scheme 1 was feasible, in which the second dehydration reaction is a rate-controlling step.

# 4. Conclusion

This work studied the self-condensation reaction of cyclohexanone catalysed by HRF5015. The effects of reaction time and temperature on the yield of dimer were investigated. HRF5015 exhibits excellent catalytic activity, and catalytic reaction can occur at 50°C. In particular, the product contains only dimers even at 100°C under the studied conditions, and the selectivity of the dimer was close to 100%. High-efficiency catalytic activity and the superior selectivity of HRF5015 may be attributed to the cluster structure formed by sulfonic groups and nanopores in HRF5015. By the self-condensation reaction of cyclohexanone for dimer with HRF5015, the apparent activation energy is $54 \ kJ \ mol^{-1}$.

Data accessibility. There are no additional data to accompany this manuscript and the electronic supplementary material. All relevant datasets are within the main body of the manuscript or the electronic supplementary material.

Authors' contributions. X.P. conducted the experiment and wrote the manuscript; S.Z. and J.Z. guided the experiment; M.Z. participated in data processing; Y.C. cooperated on this project; G.S. administrated the project and amended the manuscript. All authors gave final approval for publication.

Competing interests. We declare we have no competing interests.

Funding. G.S. acknowledges the support of the Natural Science Foundation of Shandong Province (grant no ZR2017LB005).

Acknowledgements. We thank Mr. Lianjiang Wang from the School of Foreign Languages of UJN for providing language help.

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
