## [Reviewer comments · Royal Society Open Science]

Review History

RSOS-200123.R0 (Original submission)

Review form: Reviewer 1

Is the manuscript scientifically sound in its present form?

No

Are the interpretations and conclusions justified by the results?

Yes

Is the language acceptable?

Yes

Do you have any ethical concerns with this paper?

Yes

Have you any concerns about statistical analyses in this paper?

No

Recommendation?

Major revision is needed (please make suggestions in comments)

Comments to the Author(s)

In this work, the self-condensation reaction of cyclohexanone catalyzed by HRF5015 was studied. The results are interesting, and could be published. However, please consider the following points:

1. In the catalyst recycling studies, I would do some characterization of the spent catalyst, and compare the results with those of the fresh one.
2. I miss the leaching test, please provide to confirm that the reaction is really heterogeneous.
3. I would compare the catalytic activity and selectivity of this catalyst with some others.

Review form: Reviewer 2

Is the manuscript scientifically sound in its present form?

Yes

Are the interpretations and conclusions justified by the results?

Yes

Is the language acceptable?

Yes

Do you have any ethical concerns with this paper?

No

Have you any concerns about statistical analyses in this paper?

No

Recommendation?

Major revision is needed (please make suggestions in comments)

Comments to the Author(s)

I have some concerns about this manuscript, 1) the concentration of acid of HRF5015 as a perfluorosulfonic acid resin should be provided, what's the reason cause the high activity especially when compared with other resins? 2) overall, the conversion is still quite low, can it be push forward to the desired product if some specific treatment is used, such as using molecular sieves to remove the generated water?

3) there are lots of typo errors and the English should be improved.

Decision letter (RSOS-200123.R0)

05-Mar-2020

Dear Professor Sun:

Title: Highly selective self-condensation of cyclohexanone: the distinct catalytic behavior of HRF5015

Manuscript ID: RSOS-200123

The editor assigned to your manuscript has now received comments from reviewers. We would like you to revise your paper in accordance with the referee and Subject Editor suggestions which can be found below (not including confidential reports to the Editor). Please note this decision does not guarantee eventual acceptance.

Please submit your revised paper before 28-Mar-2020. Please note that the revision deadline will expire at 00.00am on this date. If we do not hear from you within this time then it will be assumed that the paper has been withdrawn. In exceptional circumstances, extensions may be possible if agreed with the Editorial Office in advance. We do not allow multiple rounds of revision so we urge you to make every effort to fully address all of the comments at this stage. If deemed necessary by the Editors, your manuscript will be sent back to one or more of the original reviewers for assessment. If the original reviewers are not available we may invite new reviewers.

RSC Associate Editor:
Comments to the Author:
(There are no comments.)

RSC Subject Editor:
Comments to the Author:
(There are no comments.)

Reviewers' Comments to Author:

Reviewer: 1

Comments to the Author(s)

In this work, the self-condensation reaction of cyclohexanone catalyzed by HRF5015 was studied. The results are interesting, and could be published. However, please consider the following points:

1. In the catalyst recycling studies, I would do some characterization of the spent catalyst, and compare the results with those of the fresh one.
2. I miss the leaching test, please provide to confirm that the reaction is really heterogeneous.
3. I would compare the catalytic activity and selectivity of this catalyst with some others.

Reviewer: 2

Comments to the Author(s)

I have some concerns about this manuscript, 1) the concentration of acid of HRF5015 as a perfluorosulfonic acid resin should be provided, what's the reason cause the high activity especially when compared with other resins? 2) overall, the conversion is still quite low, can it be push forward to the desired product if some specific treatment is used, such as using molecular sieves to remove the generated water?

3) there are lots of typo errors and the English should be improved.

Author's Response to Decision Letter for (RSOS-200123.R0)

See Appendix A.

RSOS-200123.R1 (Revision)

Review form: Reviewer 1

Is the manuscript scientifically sound in its present form?

Yes

Are the interpretations and conclusions justified by the results?

Yes

Is the language acceptable?

Yes

Do you have any ethical concerns with this paper?

No

Have you any concerns about statistical analyses in this paper?

Yes

Recommendation?

Accept with minor revision (please list in comments)

Comments to the Author(s)

The revised MS looks much better. However, it seems to me that the authors did not carry out the leaching test. Please carry out the leaching test, either the hot filtration test, or add more reactants to the liquid phase upon completion of the first run after removal of the catalyst. Experimental data are needed to confirm that the reaction is really heterogeneous.

Review form: Reviewer 2

Is the manuscript scientifically sound in its present form?

Yes

Are the interpretations and conclusions justified by the results?

Yes

Is the language acceptable?

Yes

Do you have any ethical concerns with this paper?

No

Have you any concerns about statistical analyses in this paper?

No

Recommendation?

Accept as is

Comments to the Author(s)

The authors have carefully solved the concerns raised by the reviewers, I can recommend to accept for publication as is.

Decision letter (RSOS-200123.R1)

Dear Professor Sun:

Title: Highly selective self-condensation of cyclohexanone: the distinct catalytic behavior of HRF5015

Manuscript ID: RSOS-200123.R1

The editor assigned to your paper has now received comments from reviewers. We would like you to revise your paper in accordance with the referee and Subject Editor suggestions which can be found below (not including confidential reports to the Editor). Please note this decision does not guarantee eventual acceptance.

Please submit a copy of your revised paper before 19-Jun-2020. Please note that the revision deadline will expire at 00.00am on this date. If we do not hear from you within this time then it will be assumed that the paper has been withdrawn. In exceptional circumstances, extensions may be possible if agreed with the Editorial Office in advance. We do not allow multiple rounds of revision so we urge you to make every effort to fully address all of the comments at this stage. If deemed necessary by the Editors, your manuscript will be sent back to one or more of the original reviewers for assessment. If the original reviewers are not available we may invite new reviewers.

RSC Associate Editor:
Comments to the Author:
(There are no comments.)

RSC Subject Editor:
Comments to the Author:
(There are no comments.)

Reviewers' Comments to Author:
Reviewer: 1

Comments to the Author(s)
The revised MS looks much better. However, it seems to me that the authors did not carry out the leaching test. Please carry out the leaching test, either the hot filtration test, or add more reactants

to the liquid phase upon completion of the first run after removal of the catalyst. Experimental data are needed to confirm that the reaction is really heterogeneous.

Reviewer: 2

Comments to the Author(s)

The authors have carefully solved the concerns raised by the reviewers, I can recommend to accept for publication as is.

Author's Response to Decision Letter for (RSOS-200123.R1)

See Appendix B.

RSOS-200123.R2 (Revision)

Review form: Reviewer 1

Is the manuscript scientifically sound in its present form?

Yes

Are the interpretations and conclusions justified by the results?

Yes

Is the language acceptable?

Yes

Do you have any ethical concerns with this paper?

No

Have you any concerns about statistical analyses in this paper?

No

Recommendation?

Accept as is

Comments to the Author(s)

The revised MS looks much better, and could be accepted now.

Decision letter (RSOS-200123.R2)

Dear Professor Sun:

Title: Highly selective self-condensation of cyclohexanone: the distinct catalytic behavior of HRF5015

Manuscript ID: RSOS-200123.R2

It is a pleasure to accept your manuscript in its current form for publication in Royal Society Open Science. The chemistry content of Royal Society Open Science is published in collaboration with the Royal Society of Chemistry. I apologise that this has taken much longer than usual.

RSC Associate Editor:
Comments to the Author:
(There are no comments.)

RSC Subject Editor:
Comments to the Author:
(There are no comments.)

Reviewer(s)' Comments to Author:
Reviewer: 1

Comments to the Author(s)
The revised MS looks much better, and could be accepted now.

Appendix A

Dear Dr Laura Smith,

Thank you very much for your letter. We also thank you and reviewers for your comments and suggestions on the language and structure of our manuscript. We have modified the manuscript accordingly. The detailed corrections are listed below point by point:

Reviewers' Comments to Author:

Reviewer: 1

Comments to the Author(s)

In this work, the self-condensation reaction of cyclohexanone catalyzed by HRF5015 was studied. The results are interesting, and could be published. However, please consider the following points:

1. In the catalyst recycling studies, I would do some characterization of the spent catalyst, and compare the results with those of the fresh one.

Because it is a soft polymer material, it is difficult to characterize by electron microscopy. We observed the catalyst after the reaction. Under the protection of nitrogen, the catalyst removed from the reaction liquid was still in the block shape, with little change from before the reaction. The catalytic activity remained after five times of use, indicating that the catalyst was very stable.

2. I miss the leaching test, please provide to confirm that the reaction is really heterogeneous.

The catalyst HRF5015 is a polymer resin with excellent heat resistance, chemical stability, and high mechanical strength. HRF5015 is granular and has excellent thermal stability. It can remain stable for a long time at a temperature of about 200 °C. Besides, it is insoluble in ketone compounds. HRF5015 exists in the form of solid particles before and after the reaction, so the reaction is heterogeneous.

3. I would compare the catalytic activity and selectivity of this catalyst with some others.

The reaction conditions and results in the literature on cyclohexanone condensation reactions are summarized and listed in Table 2.

Table 2 Experimental data for Self-Condensation of Cyclohexanone

Catalyst	Reaction temperature	Dimer selectivity	Reference
HRF5015	90 °C	~100%	This work
Lewatite SPC118W	142 °C	70%	11
NaOH	140 °C	94.5%	7
Alkali carbonate	130 °C	93%	21
PW ₁₂ + PMo ₁₂ /SBA-15	150 °C	90%	22

21. Fielding M. 1970 GB Patent 1327581.

22. Wang YY, Wu B, Liu CL. 2014 Catalytic synthesis of 2-(1-Cyclohexenyl) cyclohexanone by mixed heteropoly acids catalyst (PW₁₂+PMo₁₂/SBA-15). *Asian J. Chem.* 2, 7343-7347. (doi: 10.14233/ajchem.2014.16770)

As can be seen from Table 2, with HRF5015 as catalyst for the cyclohexanone condensation reaction, the reaction temperature was lower than that of other catalysts. The selectivity of chemical reactions is crucial to production. The dimer selectivity can reach nearly 100% with HRF5015. We also made the corresponding analysis in the article.

Reviewer: 2

Comments to the Author(s)

I have some concerns about this manuscript,

1) the concentration of acid of HRF5015 as a perfluorosulfonic acid resin should be provided, what's the reason cause the high activity especially when compared with other resins?

The concentration of acid of HRF5015 has been calculated according to the ion exchange capacity and written in the article.

Compared with other resins, there are two reasons for the high catalytic activity of HRF5015. One reason is its strong acidity, which is due to the introduction of the most electronegative fluorine atom in the molecule. The strong field effect and inducing effect are generated by the fluorine atom increases the acidity dramatically.

Another reason for the high efficiency of HRF5015 may be attributed to nanochannels, and the large specific surface area of nanochannels exposing more acidic sites.

2) overall, the conversion is still quite low, can it be push forward to the desired product if some specific treatment is used, such as using molecular sieves to remove the generated water?

It can be pushed forward to the desired product if some specific treatment is used, such as using molecular sieves to remove the generated water. In this paper, we focus on catalytic activity and selectivity. Because of the reversible reaction, when the water content reaches a certain concentration, the reaction will enter the equilibrium state, so the conversion rate is not very high.

We studied the process of water removal in the reaction. The reaction conversion rate was very high, and the selectivity was still kept. The process is expected to be commercialised, so it cannot be reported yet.

3) there are lots of typo errors and the English should be improved.

The language had been revised carefully. Please find the part in color in the text.

Appendix B

Dear Dr Laura Smith,

Thank you very much for your letter. We have responded the reviewer's comments. The detailed correction is listed below.

Reviewers' Comments to Author:

Reviewer: 1

Comments to the Author(s)

The revised MS looks much better. However, it seems to me that the authors did not carry out the leaching test. Please carry out the leaching test, either the hot filtration test, or add more reactants to the liquid phase upon completion of the first run after removal of the catalyst. Experimental data are needed to confirm that the reaction is really heterogeneous.

The catalyst HRF5015, a polymer resin, has excellent thermal stability, and it can be kept stable for a long time at about 200 °C. According to the reviewers' comments, we conducted a set of experiments. A certain amount of cyclohexanone and catalyst HRF5015 were added into a 100 mL three-port bottle, protected by nitrogen gas, and the reaction was conducted at 50 °C for 1 h. The device is shown in Fig. 1.

Fig. 1 Apparatus for cyclohexanone reaction

After the reaction, the product was filtered while it was hot (Fig. 2), and the transparent particle catalyst stayed on the filter paper, indicating that this is a heterogeneous catalytic reaction. The catalyst is washed with ethanol several times and then dried and can be reused. The reaction temperature selected in this experiment is 50-100 °C, and the catalyst can maintain long-term stability. Besides, it is insoluble

in ketone compounds. HRF5015 exists in the form of solid particles before and after the reaction, so the reaction is heterogeneous.

Fig. 2 Filtration device of reaction product

We have made some modifications in the manuscript and marked it in blue.